# A Novel Drug Modulator Diarylheptanoid (*t**rans*-1,7-Diphenyl-5-hydroxy-1-heptene) from *Curcuma comosa* Rhizomes for P-glycoprotein Function and Apoptosis Induction in K652/ADR Leukemic Cells

**DOI:** 10.3390/ijms23168989

**Published:** 2022-08-12

**Authors:** Natsima Viriyaadhammaa, Suwit Duangmano, Aroonchai Saiai, Montree Tungjai, Pornngarm Dejkriengkraikul, Singkome Tima, Sawitree Chiampanichayakul, Jeffrey Krise, Songyot Anuchapreeda

**Affiliations:** 1Department of Medical Technology, Faculty of Associated Medical Sciences, Chiang Mai University, Chiang Mai 50200, Thailand; 2Cancer Research Unit of Associated Medical Sciences (AMS CRU), Faculty of Associated Medical Sciences, Chiang Mai University, Chiang Mai 50200, Thailand; 3Department of Chemistry, Faculty of Science, Chiang Mai University, Chiang Mai 50200, Thailand; 4Department of Radiologic Technology, Faculty of Associated Medical Sciences, Chiang Mai University, Chiang Mai 50200, Thailand; 5Department of Biochemistry, Faculty of Medicine, Chiang Mai University, Chiang Mai 50200, Thailand; 6Center for Research and Development of Natural Products for Health, Chiang Mai University, Chiang Mai 50200, Thailand; 7Pharmaceutical Chemistry, School of Pharmacy, University of Kansas, Lawrence, KS 66045, USA

**Keywords:** *Curcuma comosa*, diarylheptanoid, *trans*-1,7-diphenyl-5-hydroxy-1-heptene, multidrug resistance, MDR modulator, antileukemia

## Abstract

*Curcuma comosa* has been used in traditional Thai medicine to treat menstrual cycle-related symptoms in women. This study aims to evaluate the diarylheptanoid drug modulator, *t**rans*-1,7-diphenyl-5-hydroxy-1-heptene (DHH), in drug-resistant K562/ADR human leukemic cells. This compound was studied due to its effects on cell cytotoxicity, multidrug resistance (MDR) phenotype, P-glycoprotein (P-gp) expression, and P-gp function. We show that DHH itself is cytotoxic towards K562/ADR cells. However, DHH did not impact P-gp expression. The impact of DHH on the MDR phenotype in the K562/ADR cells was determined by co-treatment of cells with doxorubicin (Dox) and DHH using an MTT assay. The results showed that the DHH changed the MDR phenotype in the K562/ADR cells by decreasing the IC_50_ of Dox from 51.6 to 18.2 µM. Treating the cells with a nontoxic dose of DHH increased their sensitivity to Dox in P-gp expressing drug-resistant cells. The kinetics of P-gp mediated efflux of pirarubicin (THP) was used to monitor the P-gp function. DHH was shown to suppress THP efflux and resulted in enhanced apoptosis in the K562/ADR cells. These results demonstrate that DHH is a novel drug modulator of P-gp function and induces drug accumulation in the Dox-resistant K562 leukemic cell line.

## 1. Introduction

Chemotherapy is a primary tool for the treatment of leukemia. If the cancer is found early, chemotherapy can often have a preferable outcome. However, chemotherapy-related toxicities can occur both acutely and/or chronically. Drug resistance can occur after treatment through various mechanisms, such as increased drug efflux, decreased drug uptake, alterations in drug target sites, increased drug metabolism, escape from cell cycle checkpoints, and resistance to apoptosis [1]. One of major proteins that plays a role in drug efflux is P-glycoprotein (P-gp) [2]. P-gp is a 170 kDa transmembrane protein encoded by the *MDR1* gene. It is one of the ATP-dependent membrane transport proteins. It can facilitate the cellular efflux of amino acids, organic ions, peptides, drugs, and xenobiotics [3]. P-gp is expressed in normal organs and overexpressed in patients with AML, including patients who have been newly diagnosed and those experiencing relapse. High P-gp expression is correlated with poor outcomes, including higher white blood cell count, worse chromosomal abnormalities, and shorter survival [4,5]. NFκB and CREB protein transcription factors have been reported to regulate *MDR1* gene expression in drug-resistant cells [6,7]. Previous reports have shown that one-third of AML patients expressed P-gp at diagnosis and at relapse. These AML patients often displayed increased drug resistance and P-gp expression, which has been suggested to correlate with a reduced rate of complete remission and poor prognosis in AML patients [8]. These outcomes caused by P-gp expression reduce the intracellular drug concentrations, which decreases the cytotoxicity of the chemotherapeutic agents. Mitigation of the drug resistance phenotype can occur in at least two principle ways, which are through a decrease in P-gp expression and/or an inhibition of P-gp function. Antisense oligonucleotides, small interfering RNAs (SiRNA), and artificial transcriptional factors are strategies to modulate *MDR1* gene expression. MDR1 siRNAs have been used to suppress *MDR1* gene expression in multidrug-resistant NCI/ADR-RES breast carcinoma cells which were stably transfected with hairpin siRNA vectors [9]. The inhibition of P-gp function has been performed using previously established chemical inhibitors. The first three generations of MDR modulators are quinine, verapamil, cyclosporine-A, tariquitor, PSC 833, LY335979, and GF120918. However, these modulators must be administered in high doses to reverse MDR and have been associated with adverse effects [10]. Moreover, P-gp inhibitors including verapamil, dexverapamil, and emopamil have been developed [11]. However, these P-gp inhibitors cause immunosuppressive effects, cardiovascular disease, and have other negative side effects [11].

In an effort to decrease the aforementioned side effects of MDR modulators, natural products have been widely investigated. In addition, new nontoxic substances, such as curcumin, have been isolated and tested [12]. Curcumin I (curcuminoid) from turmeric (*Curcuma longa*) has been reported to be a modulator of the MDR phenotype [12,13,14]. It has been shown to modulate the in vitro expression and function of P-gp in the multidrug-resistant human cervical carcinoma KB-V1 cell line. Furthermore, flavonoids, terpenoids, alkaloids, and coumarins from plants, marine organisms, and microorganisms can be used as MDR modulators [15]. Bisbenzyl isoquinoline alkaloid, namely tetrandrine, from the Chinese herb *Stephania tetrandra* (Han-Fang-Chi), has been used in clinical trials for its MDR reversal activity [15]. Moreover, *S. tetrandra*, emodin, ginsenoside Rg3, artemisinin, osthole, honokiol, tea polyphenols, ligustrazine, and gaoderma from Chinese herbal medicines have been studied as reversal agents for P-gp mediated multidrug resistance in tumors [16]. In the present study, diarylheptanoid, trans-1,7-diphenyl-5-hydroxy-1-heptene (DHH) from the rhizomes of *Curcuma comosa* was investigated as an MDR modulator. The structure of DHH consists of two phenyl rings linked by a seven-carbon chain with one trans double bond and one hydroxy group, as shown in Figure 1. This plant is from the genus Curcuma and has been used in traditional Thai medicine. DHH has been purified by our research group from a hexane extraction protocol. At high concentrations DHH showed cytotoxic effects in the K562, KG1a, HL60, A549, and MCF-7 cell lines [17]. In our current work the ability of DHH to induce drug accumulation and decrease P-gp function in the drug-resistant leukemic cell line (K562/ADR) was examined. The conclusions from our studies presented here establish DHH as a new MDR modulator for P-gp function. Furthermore, DHH can reverse the MDR phenotype in K562/ADR cells.

## 2. Results

### 2.1. Cytotoxicity of DHH against Drug-Resistant K562/ADR and Drug-Sensitive K562 Cells Using MTT Assay

As shown in Figure 2, we found that, as anticipated, K562/ADR cells were more resistant to the cytotoxic effects of Dox relative to the parental K562 cell line. In contrast, DHH was found to be similarly cytotoxic to both K562 and K562/ADR cell lines (see Figure 3A,C). Moreover, the total cell number was significantly decreased after treatments with DHH at IC_20_ concentrations in both K562 and K562/ADR cells, with values of 61.2 ± 1.8 and 55.6 ± 0.6 µM, respectively (Figure 3B,D). In this study, DHH suppressed cell proliferation at the IC_20_ concentrations because it did not induce cell death after treatments.

### 2.2. Effect of DHH on P-gp Protein Expression in K562/ADR Cells

Cellular P-gp protein expression and function affect drug accumulation in the cells, which has implications for cancer treatment. The suppression of P-gp expression allows for increased drug accumulation in drug-resistant cells. In this study, P-gp expression was examined after DHH treatments with the IC_20_ value (55.6 µM) in K562/ADR cells by Western blotting. The results showed no significant decrease in the P-gp expression in the K562/ADR cells at 48 h (Figure 4). Thus, this suggests that DHH did not affect the P-gp expression in K562/ADR cells. However, the demonstration that DHH decreases cell viability and proliferation of these cells, prompted us to further explore P-gp function, as we have reported below.

### 2.3. Effects of DHH on MDR Phenotype in K562/ADR Cells

The addition of DHH to Dox resulted in a decrease in the IC_50_ cytotoxicity values for Dox in the cells examined. The reduction of Dox IC_50_ with verapamil (VP), a known inhibitor of P-gp, is shown for comparison (Figure 5). Thus, this data suggests that DHH appears to function similar to verapamil in its ability to reverse the MDR phenotype.

### 2.4. Effect of DHH on Dox Accumulation in K562/ADR Cells under Fluorescence Microscope

To visually examine the effect of DHH on Dox cellular accumulation, fluorescence microscopy was employed. DHH (55.6 µM) was added to cells along with Dox at its IC_30_, IC_40_, and IC_50_ values (33.1, 40.9, and 51.6 µM) for 4 h and compared to Dox treated cells without DHH. All cells were exposed to DAPI to help identify the nucleus. Figure 6 shows the fluorescence intensity of Dox in K562/ADR cells with or without various concentrations of DHH. These images suggest that DHH increases the cellular accumulation of Dox in a dose-dependent fashion (Compare Figure 6A,B) As an important control, we show that the DHH and the vehicle itself did not contribute to Dox fluorescence (Figure 6C). Interestingly, the higher intensity of Dox was observed when the cells were simultaneously treated with DHH, thus, suggesting that Dox accumulation was promoted by DHH addition.

### 2.5. Effects of DHH on P-gp Function in K562/ADR Cells

In this study, pirarubicin (THP) was used as a model P-gp substrate instead of Dox because THP exhibited non-interfering fluorescence and had shorter accumulation times as compared with Dox. THP is an analogue of Dox. The fluorescence intensity of THP at 590 nm in cells decreased when DHH was added to the cells indicating that DHH appeared to inhibit THP efflux in K562/ADR cells. The kinetics of THP uptake in K562/ADR cells incubated with DHH at various concentrations is shown in Figure 7. The fluorescence intensity of THP decreased when the concentration of DHH increased. The C_n_ and C_i_ values were increased in K562/ADR cells incubated with 18.8, 37.6, and 56.4 µM DHH as compared with non-incubated cells (Figure 8A,B). The kinetic parameters of the THP uptake into the DHH-incubated K562/ADR cells and non-incubated cells are shown in Table 1. The V_+_ and k_+_ values were increased in K562/ADR cells following incubation with 37.6 and 56.4 µM DHH as compared with the non-incubated cells. The V_a_ and k_a_ values were decreased in the DHH-incubated K562/ADR cells in a dose-dependent manner. Figure 8C shows the ratio of the k^i^_a_/k^0^_a_ values in the DHH-incubated K562/ADR cells. The data exhibited decreases in the k^i^_a_/k^0^_a_ ratio values in incubated cells at 9.4, 18.8, 37.6, and 56.4 µM DHH as compared with the non-incubated cells. 

### 2.6. Effects of DHH on Cytotoxicity of THP (MDR Phenotype) in K562/ADR Cells

The MDR phenotype of the K562/ADR cells was subsequently examined after exposure to both THP and DHH. The co-incubation of THP with DHH (IC_20_ concentration value of 55.6 µM) resulted in a slight reduction of the THP IC_50_. The IC_50_ value of THP with DHH was 3.16 ± 0.79 µM, whereas the IC_50_ value of THP alone was 4.27 ± 1.02 µM (Figure 9). Thus, this data suggests that DHH has a slight propensity to reverse the MDR phenotype associated with the K562/ADR cells.

### 2.7. Effect of DHH on Cell Apoptosis after Reversing the MDR Phenotype

The percentages of apoptotic cells were determined following treatment of cells with THP (at IC_30_ value of 1.36, IC_40_ value of 2.15, and IC_50_ value of 4.27 µM) with or without the addition of DHH (IC_20_ of 55.6 µM) for 48 h. The inhibition concentrations were selected according to the cell viability following THP treatments, as shown in Figure 9. THP at the IC_50_ value was used as a positive control. The apoptotic cell populations (%) of co-treatment with THP (at IC_30_ and IC_40_ values) and DHH were 33.8 ± 3.8 and 42.2 ± 2.4%, respectively. Interestingly, cells co-treated with DHH showed higher apoptotic cell population than those that only received THP. However, there were no significant differences in the IC_50_ values between co-treatments (THP+DHH) and THP (Figure 10 and Figure 11).

## 3. Discussion

DHH, isolated as a colorless oil, is an active compound from *C. comosa*. DHH has been previously isolated, and its chemical structure has been characterized by other groups [18,19,20]. This study focused on its MDR reversal activity in the K562/ADR drug-resistant leukemic cell line as compared with the drug-sensitive leukemic cell line K562. The cytotoxicity of DHH was investigated by using an MTT assay. The IC_50_ value of DHH in the K562/ADR cells was 83.6 ± 2.2 µM. This concentration was previously shown not to be toxic to peripheral blood mononuclear cells (PBMCs) [21]. In a previous report, DHH, referred to as compound **2**, did not induce hemolysis in red blood cells [17]. Thus, this concentration is presumed safe for normal human cells. DHH at IC_20_ values significantly decreased the total cell number in both K562 and K562/ADR cells as compared with vehicle control. Our study is the first to report on the impact of DHH on cell viability and the total cell number in K562 and K562/ADR cells. In this study, DHH suppressed cell proliferation at the IC_20_ concentrations because it did not induce cell death after treatments (Figure 3B,D). Then, this IC_20_ concentration of DHH was further examined for its effect on P-gp expression by Western blotting.

The reported cytotoxicity of DHH towards drug-resistant cancer cells has positive implications for successful treatment of cancer. The mechanisms involved in P-gp expression and P-gp function affect drug accumulation in the cells. The suppression of P-gp expression can allow for greater drug accumulation in cells. Small interfering RNA and microRNA-298 have been previously reported to enhance intracellular accumulation of antiepileptic drugs by suppressing the P-gp expression and selectively restored sensitivity to drugs transported by P-gp. These studies indicated that RNA interference could modulate MDR in preclinical models [22,23]. However, P-gp expression was not found to be influenced by DHH treatment. Thus, DHH did not affect the P-gp expression in K562/ADR cells, while the cell viability and total cell number were decreased after DHH treatments. Hence, the P-gp function was further investigated. 

DHH has shown strong cytotoxicity in K562/ADR cells but has no effect on P-gp expression. Therefore, the effect of DHH was examined to determine its effect on Dox-induced cytotoxicity, as measured by the inhibition of cell growth (MTT assay). Verapamil was used as a positive control reversal agent drug [12]. The co-treatment of Dox with DHH (IC_20_ concentration value of 55.6 µM) resulted in a significant increase in the cytotoxicity of Dox. Thus, DHH could reverse the MDR phenotype, showing its MDR modulator activity as a reversal agent drug. Active compounds from plants have previously been reported as MDR reversal agents. For example, triterpenoids from plants have been reported to function as reversal agents of MDR [24]. Noncytotoxic taxanes synthesized from 10-deacetylbaccatin III and special hydrophobic acylating agents have shown remarkable MDR reversal activity (≤99.8%) against drug-resistant human breast cancer cells when co-administered with paclitaxel or doxorubicin [25]. In this study, Dox accumulation was investigated using a fluorescence microscope. The fluorescence images of K562/ADR cells following Dox co-treatment with DHH exhibited more Dox uptake than those without DHH. This supports the notion that the MDR modulator properties of DHH occur through increases in Dox accumulation.

The influence of DHH on P-gp function was investigated in K562/ADR cells. In this study, pirarubicin (THP) was used as a model anticancer drug. It is an anthracycline drug and an analogue of the anthracycline antineoplastic antibiotic doxorubicin. It was used as a model drug for studying drug efflux in K562/ADR cells [26]. The remarkable properties of THP are non-interference colour and shorter uptake as compared with Dox. The fluorescence intensity at 590 nm decreased when the concentration of DHH increased, indicating that THP accumulated in K562/ADR cells and was influenced by DHH. The fluorescence intensity decreased when the concentration of DHH increased, as shown in the THP uptake kinetics shown in Figure 7. The C_n_ and C_i_ values were significantly increased in K562/ADR cells incubated with 18.8, 37.6, and 56.4 µM DHH as compared with non-incubated cells. These results suggest that DHH could increase the intracellular THP concentration in K562/ADR cells. According to the kinetic parameters shown in Table 1, the V_+_ and k_+_ values were significantly increased in K562/ADR cells following incubation with 37.6 and 56.4 µM DHH as compared with the non-incubated cells. The V_a_ and k_a_ values were decreased in the DHH-incubated K562/ADR cells in a dose-dependent manner. Moreover, the data exhibited significant decreases in the k_a_^i^/k_a_^0^ ratio values in incubated cells at 9.4, 18.8, 37.6, and 56.4 µM DHH as compared with the non-incubated cells. These data indicate that DHH has the ability to inhibit P-gp function in K562/ADR cells. Taken together, this finding suggests that DHH could decrease the function of P-gp, resulting in increased intracellular THP accumulation in K562/ADR cells.

To confirm the effects of DHH on the MDR phenotype and P-gp function, the percentages of apoptotic cells were determined after cells were treated with THP (at IC_30_, IC_40_, and IC_50_ values) or co-treatment with THP (at IC_30_, IC_40_, and IC_50_ values) and DHH (IC_20_ value of 55.6 µM) for 48 h. THP at its IC_50_ value was used as a positive control. The apoptotic cell populations (%) of THP as compared with those of co-treatments were significantly different at IC_30_ and IC_40_ values of each treatment. Whereas at IC_50_ values, there was no significant difference between treatments. The results related to the prior experiment that DHH can inhibit P-gp and increase drug accumulation. Taken together, this result indicates that DHH improves the ability of THP to induce apoptosis by inhibiting P-gp function. Taken together, DHH demonstrates a novel function as an MDR modulator by reversal of the MDR phenotype in multidrug-resistant cells. Despite the aforementioned benefits of DHH, there are few drawbacks. Even though the concentration used in this study is presumably nontoxic to normal blood cells, it is likely that fairly large doses of DHH could be required to achieve in vivo concentrations of the drug examined in this work. Further pharmacokinetic/pharmacodynamic investigations in animal models and human subjects would be necessary to definitively establish the size of the dose required to achieve the effects observed in this in vitro evaluation. 

## 4. Materials and Methods

### 4.1. Materials 

K562 cells and K562/ADR cells were purchased from RIKEN Innovation Co., Ltd. Doxorubicin was from Fresenius Kabi (Bangkok, Thailand) Ltd. Curcumin was purchased from Nacalai Tesque (Kyoto, Japan), pirarubicin (4-Q-tetrahydropyrayldoxorubicin) and verapamil were purchased from Sigma-Aldrich (St. Louis, MO, USA). Rabbit monoclonal anti-P-gp IgG was purchased from Boster Biological Technology, while rabbit polyclonal anti-human GAPDH IgG anti-GAPDH was purchased from Santa Cruz Biotechnology, Santa Cruz, CA, USA. The apoptosis kit was BioLegend (Santa Cruz, CA, USA).

### 4.2. DHH Purification

DHH was purified as previously described [17]. Briefly, rhizomes of *Curcuma comosa* were peeled and ground into a powder. Then, the powder was macerated in *n*-hexane to obtain a partially purified fraction of hexane. Column chromatography was performed to isolate the main compound from the partially purified fraction of hexane. The chemical structure of the main compound was determined using the nuclear magnetic resonance (NMR) technique. The diarylheptanoid, *t**rans*-1,7-diphenyl-5-hydroxy-1-heptene (DHH), was found as the main compound of the partially purified fraction of hexane.

### 4.3. Cell Culture

Leukemic cell lines including K562 and K562/ADR were maintained in the RPMI 1640 medium containing 1 mM l-glutamine, 100 units/mL penicillin, and 100 µg/mL streptomycin, and supplemented with 10% fetal bovine serum (FBS). Doxorubicin was used monthly to maintain the drug-resistant phenotype of K562/ADR at the final concentration of 1 µM. The cancer cell lines were both incubated at 37 °C under 95% humidity and 5% CO_2_.

### 4.4. Cytotoxicity and Doxorubicin Resistance Reversal Determination Using MTT Assay

The MTT assay used MTT (3-(4,5-dimethylthiazol-2-yl)-2,5-diphenyltetrazolium bromide) to evaluate the cell viability. Viable cells can reduce the MTT dye to purple formazan crystal, indicating cell viability. The cytotoxicity of the active compounds and drugs were investigated in the K562 and K562/ADR cells. K562 (1.0 × 10^4^ cells/well) and K562/ADR (1.0 × 10^4^ cells/well) were treated with DHH or drugs in 96-well plates and incubated for 48 h at 37 °C under 95% humidified and 5% CO_2_. Then, 100 µL of cell supernatant were removed and added to 15 µL of the MTT dye, and cells were further incubated for 4 h. Following the supernatant discarding, 200 µL of DMSO were added to dissolve the purple formazan crystal and mixed well. The optical density was measured using an ELISA plate reader at 578 nm with a reference wavelength at 630 nm. The percentage of viable cells was calculated from the absorbance values of the test and control wells using the following equation: (1)$$\begin{array}{cc} \%\;\mathrm{Cell}\;\mathrm{viability}= & \frac{\mathrm{Mean}\;\mathrm{absorbance}\;\mathrm{in}\;\mathrm{test}\;\mathrm{well}}{\mathrm{Mean}\;\mathrm{absorbance}\;\mathrm{in}\;\mathrm{vehicle}\;\mathrm{control}\;\mathrm{well}} \end{array}\times 100$$

The average percentage of surviving cells at each concentration, obtained from the triplicate experiments, was plotted as a dose-response curve. The 50% inhibitory concentration (IC_50_) was defined as the lowest concentration that inhibited cell growth by 50% as compared with the untreated control.

### 4.5. Trypan Blue Exclusion Test

Viable cells with intact membranes can exclude trypan blue; however, dead cells with damaged membrane are stained with trypan blue dye. To perform the trypan blue exclusion test, the trypan blue dye solution (0.2%) and cell suspension were mixed. Then, the viable (unstained) and dead (stained) cells were counted using the hemacytometer under a microscope. The percentages of both viable and dead cells were calculated.

### 4.6. Western Blot Analysis

The K562/ADR (1.0 × 10^5^ cells/mL) cells were treated with DHH for 48 h at 20% growth inhibition (IC_20_). The IC_20_ values were used for the treatment to determine the compound effects on the P-gp expression in cells without the interference of dead cells. After the treatments, the cells were harvested and washed with PBS, pH 7.2, three times. The whole protein lysates were extracted using the RIPA buffer (50 mM Tris-HCl, 150 mM NaCl, 1% Triton X-100, 0.5 mM EDTA, 0.1% SDS, and a protease inhibitor cocktails). The protein concentration was determined using the Folin–Lowry method. Then, 50 µg each sample were loaded to 7.5% SDS-PAGE and transferred to PVDF membranes. After transferring, the membranes were shaken in PBS, pH 7.4, 5 min. Then, the membranes were blocked in 5% skim milk in PBS, pH 7.4, overnight at 4 °C. Each membrane was incubated at room temperature (25 °C) with rabbit monoclonal anti-P-gp IgG (Boster Biological Technology, Pleasanton, CA, USA) and rabbit polyclonal anti-human GAPDH IgG (Santa Cruz Biotechnology, Santa Cruz, CA, USA) at a dilution of 1:1000 while shaking for 2 h and 1 h, respectively. Membranes were subsequently washed with 0.1% Tween 20^®^ and HRP-conjugated goat anti-rabbit IgG (Invitrogen™, Massachusetts, CA, USA) at 1:20,000. The dilution was then added and incubated for 2 h at room temperature (25 °C). To detect protein bands, Luminata™ Forte Western HRP substrate (Merck, Darmstadt, Germany) was added to the membranes. Then, the membranes were placed onto a film cassette and exposed to an X-ray film (Sakura, Osaka, Japan). Densitometry was quantitated using the Quantity One 1D Analysis software (Bio-Rad, Hercules, CA, USA). The density values of the P-gp bands were normalized to the GAPDH bands.

### 4.7. Drug Accumulation in K562/ADR Cells by Fluorescence Microscope

The K562/ADR cells (1.0 × 10^5^ cells/mL) were treated with Dox or Dox co-treatment with DHH (IC_20_ value of 55.6 µM). Dox was prepared in various concentrations (IC_30_–IC_50_). After 4 h, medium was removed and cells were washed with PBS, pH 7.2, two times. Cells were fixed with 2% paraformaldehyde in Hanks’ balanced salt solution for 20 min. Then, the fixed cells were washed with PBS, pH 7.2. DAPI (Sigma, St. Louis, MO, USA) at the concentration of 2 µg/mL was used to stain nucleus for 10 min. Cell smear was prepared and mounted with VECTASHIELD^®^ antifade mounting medium (Vector Laboratories, Newark, CA, USA) on a slide and sealed. The slide was visualized under a 3I/Olympus epifluorescence inverted microscope (Olympus, Center Valley, PA, USA).

### 4.8. Determination of P-gp Function in Living K562/ADR Cells

The P-gp function in living K562/ADR cells was evaluated using a non-invasive functional spectrofluorometric method (LS55; Perkin Elmer, Inc., Massachusetts, CA, USA), as previously described [27]. Briefly, the K562/ADR cells (2.0 × 10^6^ cells) were incubated with DHH at various concentrations (9.4, 18.8, 37.6, and 56.4 µM) in 2 mL of Lukholff-Na^+^ buffer, pH 7.2, in 1 cm quartz cuvettes, and vigorously stirred at 37 °C. Glucose (5 mM) was added as the energy supplement. THP (1 μM) was added to the system. The fluorescence intensity of the THP at 590 nm (excitation at 480 nm) was monitored as function of time. A decrease in the fluorescence intensity was observed after the THP intercalated between the base pairs of DNA. At the steady state, the cells were permeabilized with 0.01% Triton X-100 and entered the equilibrium state. The value of the overall nuclear drug concentration at the steady state (C_n_) and the intracellular drug concentration (C_i_) were calculated. 

In the current study, the P-gp function was indicated by the P-gp-mediated active efflux coefficient (k_a_). The inhibitory ability of DHH on the P-gp function was determined using the ratio: k^i^_a_/k^0^_a_ (k^i^_a_, P-gp mediated active efflux of THP in the presence of DHH and k^0^_a_, P-gp mediated active efflux of THP in the absence of DHH). The ratio k^i^_a_/k^0^_a_ was equal to 0 if the P-gp mediated active efflux was completely inhibited. The ratio k^i^_a_/k^0^_a_ was equal to 1 if the P-gp mediated active efflux was not inhibited. 

The parameters, including the rate of THP passive uptake (V_+_), passive influx coefficient (k_+_), rate of P-gp mediated active efflux of THP (V_a_), and active P-gp mediated drug efflux coefficient (k_a_), were also determined.

### 4.9. Effect of DHH on the Cytotoxicity of THP (MDR Phenotype) in K562/ADR Cells

To study the effect of DHH on the cytotoxicity of Dox, K562/ADR cells (1.0 × 10^4^ cells/well), in 100 µL medium, were cultured in the presence of 0.5% DMSO (vehicle control) or 55.6 µM DHH in the presence of different concentrations of THP (0.2–100 μM). The viable cells were determined using the MTT assay.

### 4.10. Apoptosis Induction in K562/ADR Cells

The K562/ADR (1.0 × 10^5^ cells/mL) cells were treated with THP or THP combined with DHH (IC_20_ value of 55.6 µM) for 48 h at the IC_30_ value of 1.36, the IC_40_ value of 2.15, and the IC_50_ value of 4.27 µM. THP at the IC_50_ value was used as a positive control. After 48 h, cells were harvested and washed with cold PBS, pH 7.2. PI and Annexin V-FITC were added according to manufacturer (BioLegend, CA, USA). Cells were gently vortexed and incubated for 15 min at 25 °C in a dark place. Then, buffer was added to each tube and analyzed with flow cytometer. The results were analyzed using the FlowJo V10 program (Ashland, OR, USA). The positive control was used to set up the quadrants for cell population determination.

### 4.11. Statistical Analysis

The data are expressed as the mean ± standard deviation (SD) from the triplicate samples of three independent experiments. The statistical differences between the means were determined using one-way ANOVA and Student’s *t*-test. The differences were considered significant when the obtained probability value was found to be less than 0.05 (*p* < 0.05).

## 5. Conclusions

This study used the identified diarylheptanoid, *trans*-1,7-Diphenyl-5-hydroxy-1-heptene (DHH), against the multidrug-resistant leukemic K562/ADR cells. DHH alone significantly decreased the cell viabilities and total cell number. When combined with doxorubicin, DHH was discovered to be a novel drug modulator. The co-treatment of doxorubicin and DHH showed a decrease in cell viability as compared with treatment without DHH. DHH is a potent agent to significantly decrease P-gp function and induce drug accumulation. In addition, DHH does not induce hemolysis and is not toxic to peripheral blood mononuclear cells at the IC_20_ value, which could be an advantage over other drug modulators. These results suggest that DHH has chemosensitizer potential. Further investigations should be performed to develop new agents for drug-resistant leukemia treatment.

## Figures and Tables

**Figure 1 ijms-23-08989-f001:**
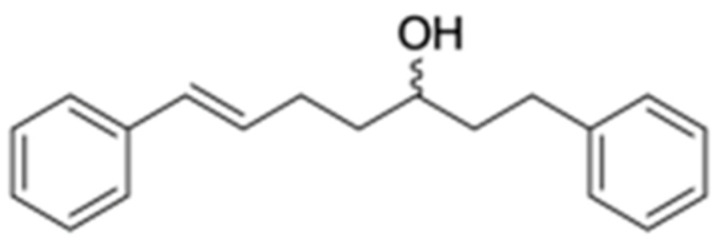
The chemical structure of DHH (*trans*-1,7-diphenyl-5-hydroxy-1-heptene).

**Figure 2 ijms-23-08989-f002:**
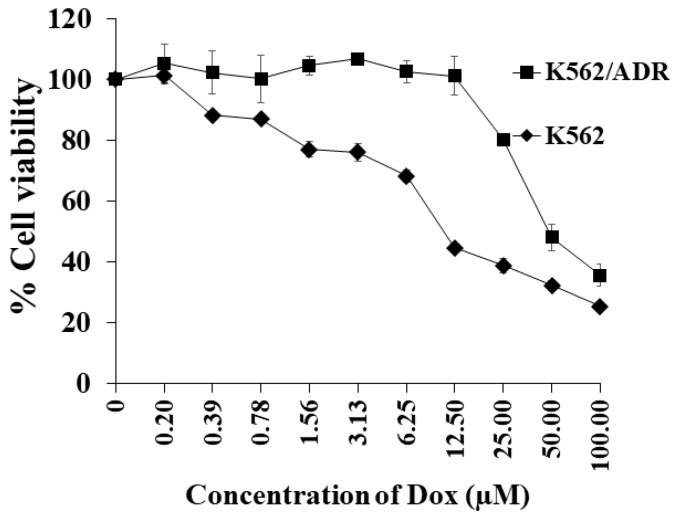
Effects of doxorubicin (Dox) on the cytotoxicity in drug-resistant K562/ADR and drug-sensitive K562 cells.

**Figure 3 ijms-23-08989-f003:**
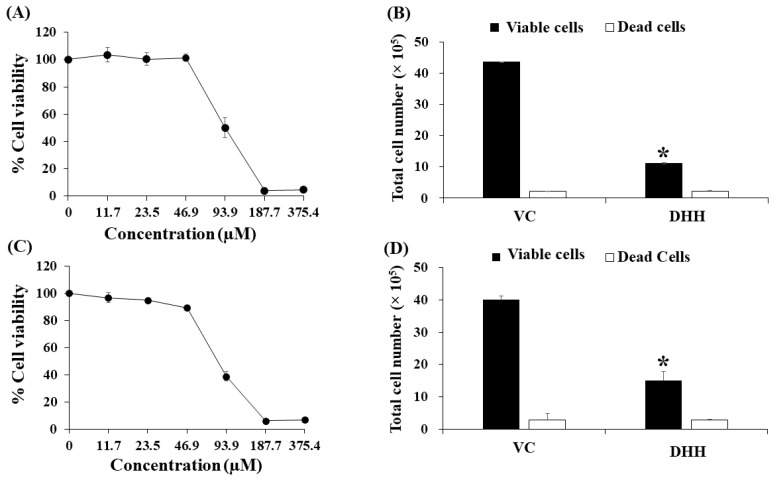
Effects of DHH from *C. comosa* on the cytotoxicity and total cell number of drug-resistant K562/Dox and drug-sensitive K562 cells. K562/ADR and K562 cells were treated with DHH for 48 h: (**A**) Cytotoxicity of DHH treatment in drug-sensitive K562 cells by MTT assay; (**B**) total number of K562 cells after DHH treatment at IC_20_ value (61.2 µM) was determined using the trypan blue exclusion method; (**C**) cytotoxicity of DHH in drug-resistant K562/ADR cells by MTT assay; (**D**) total cell number of K562/ADR cells after DHH treatment at IC_20_ value (55.6 µM) was determined using the trypan blue exclusion method. The data are shown as mean ± SD from independent experiments performed in triplicate. The significance of mean differences was assessed using one-way ANOVA. * *p* < 0.05 as compared with vehicle control (VC).

**Figure 4 ijms-23-08989-f004:**
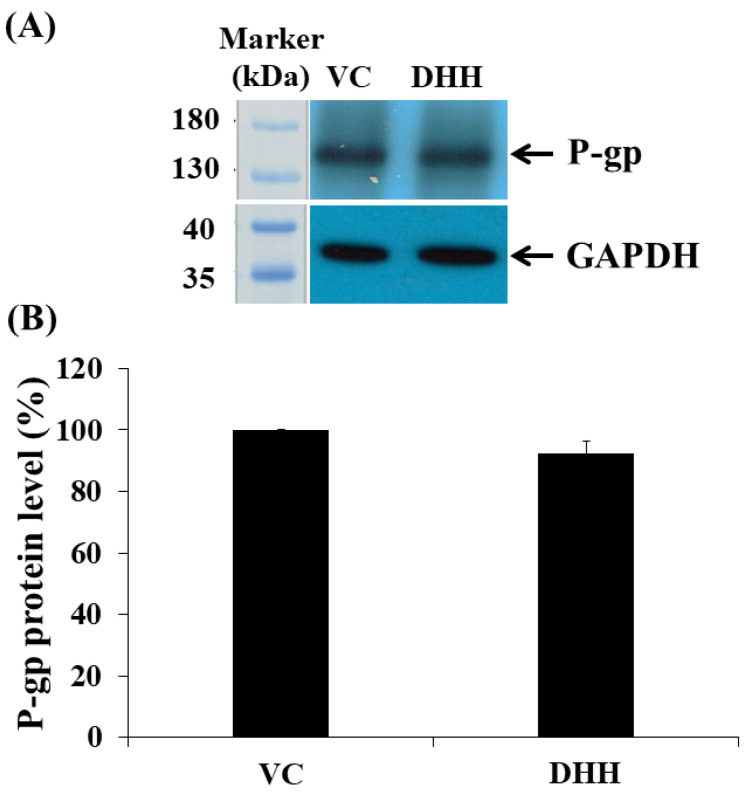
Effect of DHH on the P-gp expression in K562/ADR cells. K562/ADR cells were cultured with DHH at the IC_20_ value (55.6 µM) for 48 h. The P-gp level was determined by Western blotting using (**A**) rabbit monoclonal anti-P-gp IgG and (**B**) quantified by densitometry. Glyceraldehyde phosphate dehydrogenase (GAPDH) was used to normalize the level of P-gp. Data represent mean ± SD of three dependent experiments. The significance of mean differences was assessed using one-way ANOVA.

**Figure 5 ijms-23-08989-f005:**
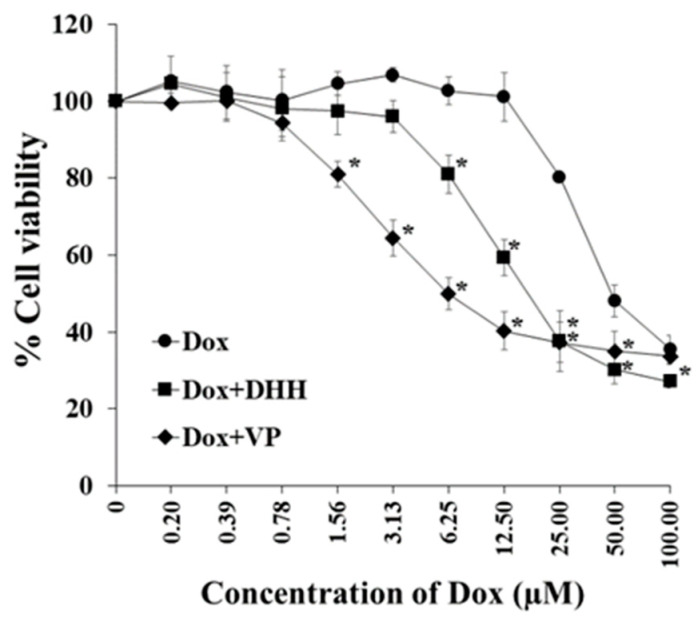
Effect of DHH on the cytotoxicity of Dox. K562/ADR cells (1.0 × 10^4^ cells/well) were cultured in 100 µL medium in the presence of 0.5% DMSO, with 22 µM verapamil (VP) or 55.6 µM DHH in the presence of different concentrations of Dox. The viable cells were determined using an MTT assay. The data are shown as mean ± SD from 3 independent experiments performed in triplicate. The significance of mean differences was assessed using one-way ANOVA. * *p* < 0.05 as compared with Dox treatment.

**Figure 6 ijms-23-08989-f006:**
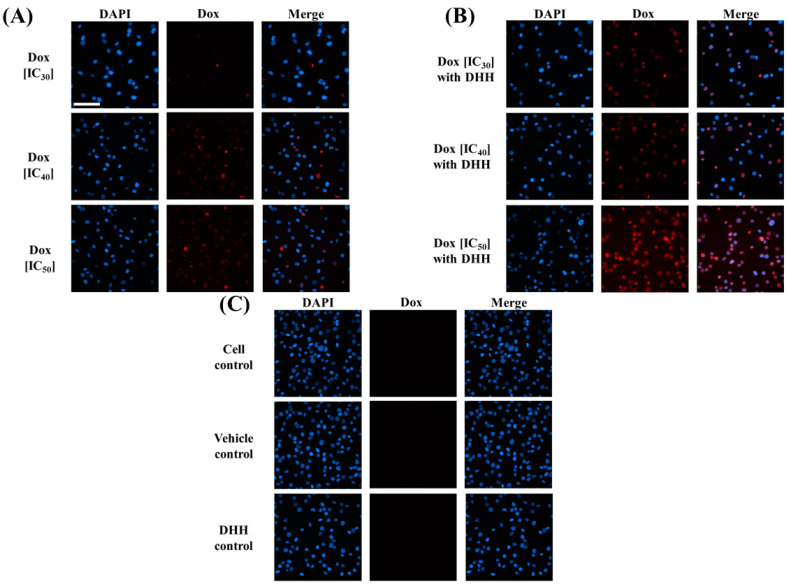
Dox accumulation in K562/ADR cells under fluorescence microscope (scale bar, 100 µm): (**A**) K562/ADR cells were treated with Dox at various concentrations; (**B**) K562/ADR cells were cotreated with Dox at various concentrations (33.1, 40.9, and 51.6 µM) and DHH (55.6 µM); (**C**) cell control, vehicle control, and DHH control showed no autofluorescence interference to DOX fluorescence intensity.

**Figure 7 ijms-23-08989-f007:**
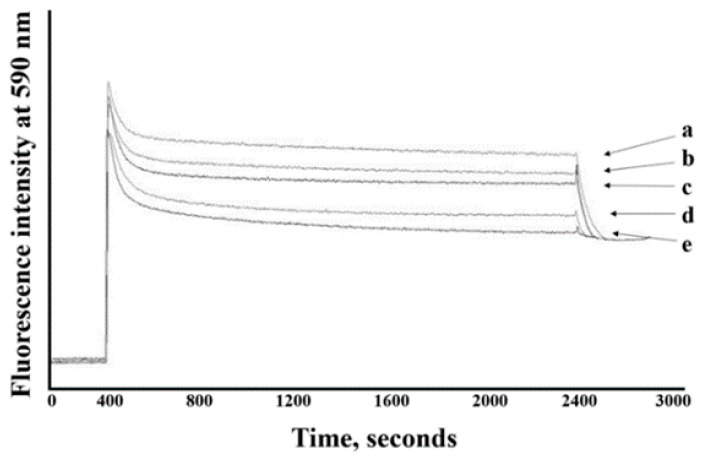
Effect of DHH on the pirarubicin (THP) accumulation in K562/ADR cells. a, vehicle control; b, 9.4 µM; c, 18.8 µM; d, 37.6 µM; and e, 56.4 µM.

**Figure 8 ijms-23-08989-f008:**
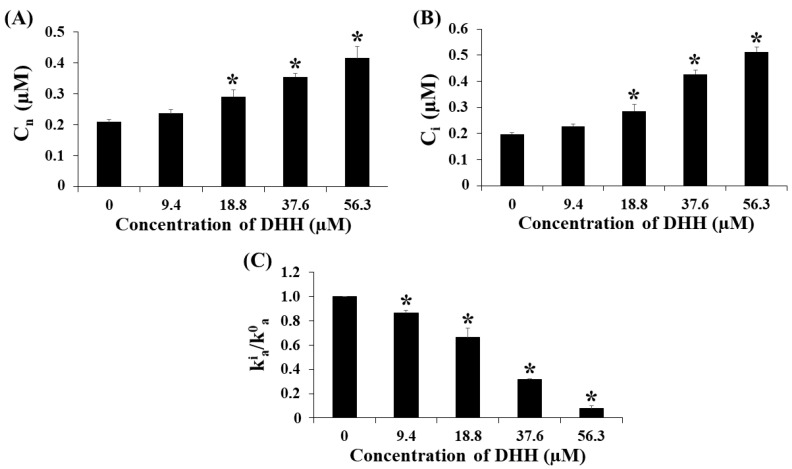
Ability of DHH to inhibit the P-gp mediated efflux of the drug in K562/ADR cells. Experiments were conducted in 1 cm quartz cuvettes under vigorous stirring. Cells were incubated with 5 mM glucose, 1 µM of pirarubicin (THP), 0.01% Triton X-100, and various concentrations of DHH: (**A**) Overall nuclear concentration in the steady state (C_n_) of THP after the K562/ADR cells were incubated with various concentration of DHH; (**B**) the intracellular concentration of pirarubicin (C_i_) after the K562/ADR cells were incubated with various concentrations of DHH; (**C**) the k^i^_a_/k^0^_a_ ratio, representing a decrease in the P-gp mediated THP efflux. The data are shown as mean ± SD from 3 independent experiments performed in triplicate. The significance of mean differences was assessed using one-way ANOVA. * *p* < 0.05 as compared with a condition without DHH.

**Figure 9 ijms-23-08989-f009:**
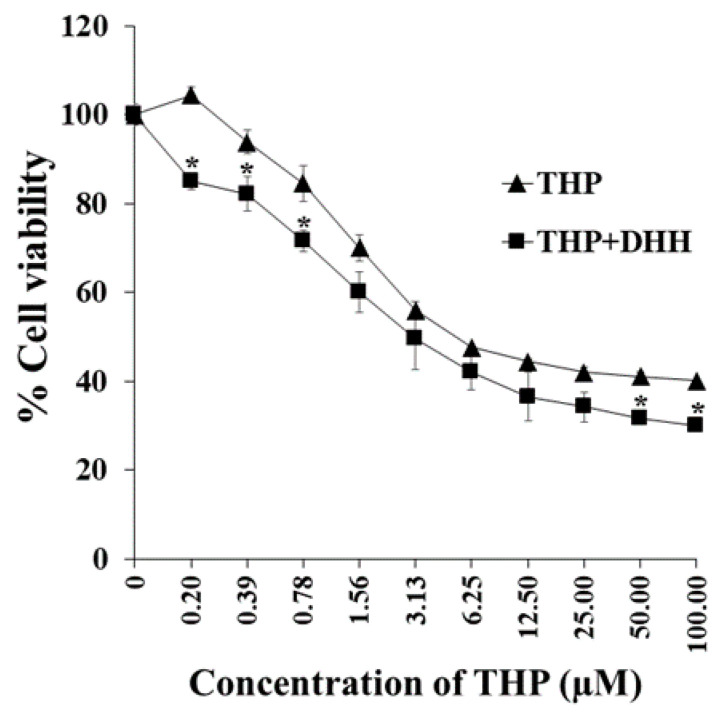
Effect of DHH on the cytotoxicity of THP. K562/ADR cells (1.0 × 10^4^ cells/well) were cultured in 100 µL medium in the presence of 0.5% DMSO, with 55.6 µM DHH in the presence of different concentrations of THP (0.2–100 µM). The viable cells were determined using an MTT assay. The data are shown as mean ± SD from independent experiments performed in triplicate. The significance of mean differences was assessed using one-way ANOVA. * *p* < 0.05 as compared with THP treatment.

**Figure 10 ijms-23-08989-f010:**
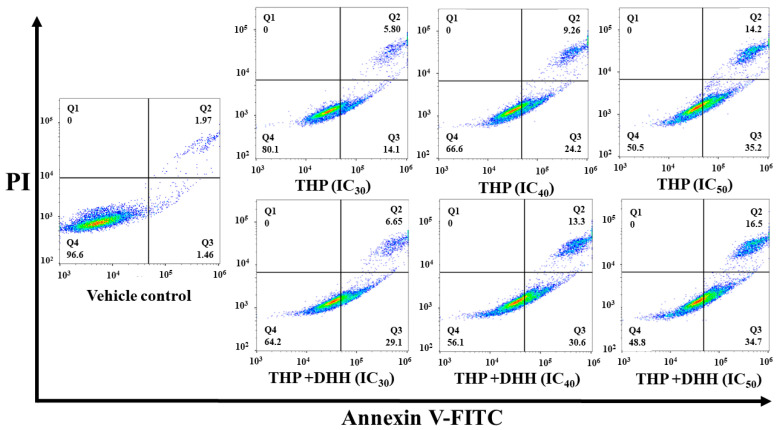
Effects of THP and THP combined with DHH on apoptosis induction in K562/ADR cells. K562/ADR cells (1.0 × 10^5^ cells/mL) were cultured with various concentration of THP or 55.6 µM of DHH in the presence of different concentrations of THP. Apoptotic cells were determined by staining with PI and Annexin V-FITC. The representative flow cytometry dot plot indicating cell population (Q1, necrotic cells; Q2, apoptotic cells; Q3, early apoptotic cells; Q4, live cells).

**Figure 11 ijms-23-08989-f011:**
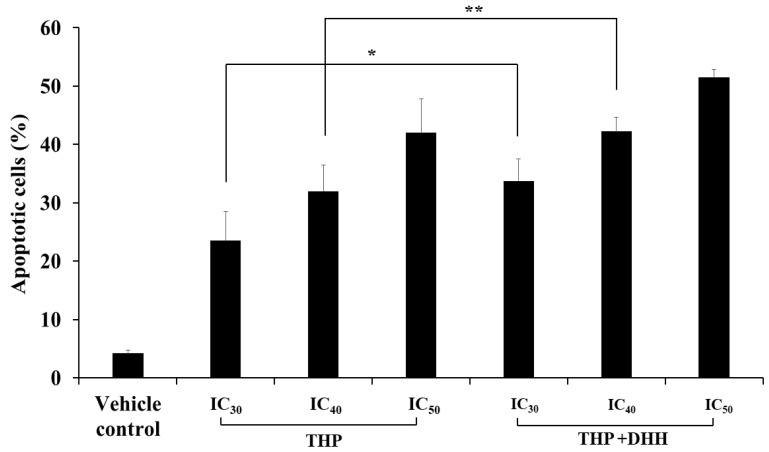
Effects of THP and THP combined with DHH on apoptosis induction in K562/ADR cells. K562/ADR cells (1.0 × 10^5^ cells/mL) were cultured with various concentrations of THP or 55.6 µM of DHH in the presence of different concentrations of THP. Apoptotic cells were determined by staining with PI and Annexin V-FITC. The percentage of apoptotic cells were compared. The data are shown as mean ± SD from three independent experiment. The significance of mean differences was assessed using one-way ANOVA. * *p* < 0.05 as compared with THP treatment at IC_30_. ** *p* < 0.01 as compared with THP treatment at IC_40_.

**Table 1 ijms-23-08989-t001:** Kinetics of the pirarubicin uptake into K562/ADR after treatment with various concentrations of DHH.

DHH, µM	V_+_, nM/s	K_+_ × 10^−9^ L/cell.s	V_a_, nM/s	K_a_ × 10^−9^ L/cell.s
0	0.73 ± 0.1	0.73 ± 0.02	0.56 ± 0.1	2.62 ± 0.28
9.4	1.06 ± 0.1	1.06 ± 0.13	0.54 ± 0.1	2.50 ± 0.27
18.8	1.01 ± 0.2	1.01 ± 0.15	0.49 ± 0.1	2.06 ± 0.21
37.6	1.23 ± 0.1 ^a^	1.23 ± 0.10 ^a^	0.37 ± 0.1 ^a^	0.78 ± 0.14 ^a^
56.3	1.26 ± 0.2 ^a^	1.26 ± 0.18 ^a^	0.11 ± 0.1 ^a^	0.21 ± 0.04 ^a^

Results expressed as mean ± SD of triplicate samples. V_+_, rate of pirarubicin uptake; K_+_, passive influx coefficient; V_a_, rate of P-gp mediated active efflux of pirarubicin; K_a_, P-gp mediated active efflux coefficient. Superscript letters (^a^ *p* < 0.05 as compared with control) within the same column denote significant differences in the means between the different samples determined using one-way analysis of variance (ANOVA), followed by Tukey’s test (*p* < 0.05).

## Data Availability

Not applicable.

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
