# Peer review of "A Novel Drug Modulator Diarylheptanoid (trans-1,7-Diphenyl-5-hydroxy-1-heptene) from Curcuma comosa Rhizomes for P-glycoprotein Function and Apoptosis Induction in K652/ADR Leukemic Cells"

_ijms, 2022, doi:10.3390/ijms23168989_

Round 1

Reviewer 1 Report

The present paper investigates the potential use of a novel drug  derived from Curcuma comosa to overcome MDR resistance in K562/ADR leukemic cells.

Although data demonstrating the potential utilization of this novel drug appear of interest the paper needs to be better structured and extensively revised. Some paragraphs look unclear requiring a more clear and detailed explanation.

-          First of all the Authors must revise the English form  throughout the manuscript.  Several errors are present , see for example line 37: “showed a suppress” “showed to suppress”; lines 87-95: “in this study”, “In the present study”………..lines 93-94” in the drug -resistant leukemic cell line  is repeated two times; line 95: “this is the first report to present” This is the first report that present….)

-           

-          Paragraph 2.1:

-          Please introduce figure 1 and the chemical structure of DHH

-          In my opinion paragraph 2.1 shoud be rewritten : i.e. “   As shown in Figure 2 we confirmed the drug resistant phenotype of  the Adryamicin -resistant K562 leukemic cell line (K562/ADR) . K562/ADR cells in fact exhibited enhanced resistance to doxorobucin treatment in comparison with the drug-sensitive K562 cell line, as determined by MTT assay (Figures………). When we further tested DHH in the K562/ADR cell line we observed a strong cytotoxicity with a IC50 of ……….. after 48h of incubation………………

-          Delete lines 112-114.

-          Figure 3B and 3C, please insert what means VC

-          In my opinion  Figure 3 should be inserted just below paragraph 2.1 and Figure 4 below paragraph 2.2.

-          Paragraph 2.2 :

-          I think that the Authors may introduce here that they have further examined the effect of DHH IC20 concentration by western blotting in K562/ADR cells.

-          Line 140 : The results showed no significant decrease …………. , thus suggesting that DHH did not affect the P-gp expression in K562/ADR cells.  However, the demonstration  that DHH decreases  cell viability and proliferation of these cells,  prompted us to  further explore P-gp function, as we have reported  below.  (….may be this is better?’)

-          Paragraph 2.3:

      Lines 217-219 : please rewrite explaining better the treatments: Increased concentration of DOX with/without co-treatment with DHH.

-          Lines 221-223: I think needs to be rewritten :  Interestingly the higher intensity of DOX was observed when the cells were contemporarely treated with DHH, thus suggesting that DOX accumulation was promoted by DHH addition.

-          Paragraph 2.6:  Delete lines 270-271: The Authors have already specified why they use THP in place of DOX ( see lines 226-227) .    We then addressed cytotoxicity and apoptosis induction by THP or THP +DHH in K562/ADR cells. Please rewrite paragraphs 2.6 and 2.7: they are actually very confusing. Co-incubation?? Co-treatment……  Enhanced cytotoxicity was detected  when K562/ADR cells were co-treated with DHH and THP” .

-          In Paragraph 2.7 The Authors have again repeated why they have chosen to use THP!!!

-          DISCUSSION :

-          Lines 307-309. The Authors say that DHH induces suppression but not cell death of leukemic cells. It could be of help to show cell cycle of K562/ADR cells after DHH treatment.

-           

-          As a general comment the Authors have to rewrite all the paper trying to better describe the results avoiding any repetitions. Also in the Discussion they have to better emphasize the novelty of this research by providing a clear message.

Author Response

Dear Reviewer #1,

Thank you very much for your kind comments and suggestions. Your questions and suggestions are valuable for our paper. We have already answered your questions and suggestions as attached file. Moreover, the English form have been revised throughout the manuscript with red color of track changes by native speaker (Prof.Dr.Jeffrey Krise).

Reviewer 2 Report

the paper is well-written with an interesting research topic. the authors explain the methodology precisely with enough information to repeat experiments. the discussion mainly focuses on all the benefits of the investigated substance. the conclusion can be easily drawn from the discussion. I suggest to authors just to briefly explain what are misfalls of DHHa.  

Author Response

Dear Reviewer #2,

Thank you very much for your kind comments and suggestions. Your questions and suggestions are valuable for our paper. We have already answered your questions and suggestions as attached file. 

Round 2

Reviewer 1 Report

The manuscript has been extensively revised. This version appears more clear in terms of the data shown and the English language results extensively improved. The Authors have therefore adequately addressed all the points raised.